# Niche Evolution and Conservation of a Chinese Endemic Genus *Sinojackia* (Styracaceae)

**DOI:** 10.3390/biology13121085

**Published:** 2024-12-22

**Authors:** Miao Feng, Jisi Zhang

**Affiliations:** Liaoning Key Laboratory of Development and Utilization for Natural Products Active Molecules, Anshan Normal University, Anshan 114000, China; fengmiao@asnc.edu.cn

**Keywords:** *Sinojackia*, conservation, niche evolution, phylogeny, divergence time

## Abstract

The climate changes resulted in the expansion and/or contraction of plants and the fragmented distribution of certain species. Here, we predicted the potential distribution areas of a Chinese endemic genus *Sinojackia*, and our results showed that: The monophyletic genus *Sinojackia* originated at middle Miocene and diverged at the late Miocene. The aridity index had the highest contribution for the niche evolution of *Sinojackia*. The precipitation of the warmest quarter was the main environmental variable for the distribution of *Sinojackia* in the Last Glacial Maximum (LGM), while the human footprint is main variable in the current era and 2081–2100. Compared to the highly suitable distribution area in the LGM, the genus *Sinojackia* would expand during the near-current era and 2081–2100.

## 1. Introduction

It is well-known that climate change is expected to have a profound impact on the species’ distribution, such as the temperature since the Late Miocene experiencing a continuous decline [1], which led to dramatic biodiversity changes [2,3]. The temperature changes resulted in the expansion and/or contraction of plants and the fragmented distribution pattern of certain species [4,5]. Evergreen broad-leaved forests (EBLFs) largely occur in East Asia under a monsoon climate with obvious seasonality, which comprise the main portion of the global subtropical biodiversity [6,7]. East Asian EBLFs harbor numerous relicts or endemic species [6,8], and these forests are widely regarded as biodiversity museums or refugia [9]. Understanding climate change and the species distribution range is of great significance to the preservation of threatened and endemic species in this biome.

*Sinojackia* Hu (Styracaceae) contains five species, i.e., *S. xylocarpa* Hu, *S. sarcocarpa* L.Q. Luo, *S. rehderiana* Hu, *S. heneyi* (Dummer) Merrill, and *S. dolichocarpa* C.J. Qi [10]. Recently, there are three new species (*S. huangmeiensis* J.W. Ge & X.H. Yao, *S. microcarpa* C.T. Chen & G.Y. Li, and *S. oblongicarpa* C.T. Chen & T.R. Cao), and one new subspecies (*S. xylocarpa* var. *leshanensis* L.Q. Luo) has been established [11,12,13,14]. In total, there are eight species and one subspecies recorded in *Sinojackia*. This genus is endemically distributed in Central, Southern, and Southwest China [10,15]. All *Sinojackia* species have been listed in National Key Protected Wild Plants of China as Grade II (2021) owing to their small population sizes and poor recruitment within populations. Especially, *S*. *xylocarpa* is assessed as Vulnerable (VU) in the IUCN Red List of Threatened Species (www.iucnredlist.org/species/32374/9701730, accessed on 1 February 2024). These species can always be found on mountain slopes or along stream banks at an altitude of 20–3500 m [10,11,12,13,14]. Yao et al. [15] conducted extensive fieldwork on *Sinojackia* and they found that most species of *Sinojackia* maintained a small number of populations with a very small population size, and some species located in natural reserves. About six species have been ex situ conserved in botanical gardens, such as the Wuhan Botanical Garden (China) and C.R. Parks residence (USA) [15,16,17]. Furthermore, *Sinojackia* species are deciduous trees or shrubs with an elongated pedicle, campanulate corolla, and elongated style, but the eight species are delimitated substantially on their weighing-hammer-like fruit morphology [10,11,12,13,14]. *Sinojackia* species have high gardening-plant values, because they have white fragrant flowers and unique weighing-hammer-like fruits [16]. Therefore, assessing the distribution status of *Sinojackia* has important significance for the conservation of *Sinojackia*, especially considering future climate scenarios.

Until now, many works of research about phylogeny and biogeography have been conducted, and all of them supported the idea that *Sinojackia* is monophyletic with moderate to high supporting values [17,18,19]. For example, Fritsch et al. [17] reconstructed the phylogeny of the Styracaceae based on 3 DNA markers (*trnL*-*L*-*F*, *rbcL*, and ITS) and 47 morphological characters, and showed that *Changiostyrax dolichocarpa* (previously, *S. dolichocarpa*) did not cluster with *Sinojackia*. Yao et al. [18] combined ITS, *psbA*–*trnH*, and microsatellite data to infer the phylogeny of *Sinojackia*, and supported *S. dolichocarpa* as a member of *Changiostyrax*. However, the results of the two studies have not obtained strong supporting values for the monophyly of *Sinojackia*. Recently, Jian et al. [19] supported well the idea that the genus *Sinojackia* was a monophyletic group and divided it into two clades. They also showed that *Sinojackia* originated in Central–Southeast China during the early Miocene, and the glacial–interglacial interactions with the monsoon climate may provide to be favorable expansion conditions for *Sinojackia*. However, there are few studies for predicting the impact of future climate change on *Sinojackia* species’ distributions.

Ecological niche modeling is widely used to predict the potential distribution area of species under current and future climate scenarios. For instance, Xiao et al. [20] identified the pivotal role of precipitation for affecting the distribution of *Tsuga*, and highlighted the conservation of natural *Tsuga* distributions in East Asia and North America. Qiu et al. [21] revealed that the geographical distribution of the three *Bergenia* (Saxifragaceae) species was primarily influenced by precipitation and elevation. By 2090, the three *Bergenia* species were expected to show contrasting range changes, and they were projected to shift their ranges to higher elevations in response to temperature increases. Consequently, these studies provide valuable information for biodiversity conservation and management.

In this study, we applied the MaxEnt model to predict the potential distribution of *Sinojackia* under current and future climate change scenarios. Specifically, the objectives are as follows: (1) to explore the niche evolution of *Sinojackia* and the critical environmental factors; (2) to assess the effects of climate change and the human footprint on the geographic range changes of *Sinojackia* from the LGM and the future; and (3) to propose a conservation strategy for *Sinojackia*. This study will provide valuable insights for the conservation of the endemic plants under climate change.

## 2. Materials and Methods

### 2.1. Taxon Sampling

A total of 14 individuals representing six *Sinojackia* species were sampled, including three individuals of *S. oblongicarpa*, three individuals of *S. xylocarpa*, three individuals of *S. sarcocarpa*, and three individuals of *S. microcarpa*. There are seven species of Styraxaceae selected as outgroups, including *Changiostyrax dolichocarpus*, *Halesia diptera*, *Melliodendron xylocarpum*, *Perkinsiodendron macgregorii*, *Pterostyrax hispidus*, *Rehderodendron macrocarpum*, and *Styrax wuyuanensis*. All plastome sequences were downloaded from GenBank (Appendix A).

### 2.2. Phylogeny Reconstruction

Sequence alignments were performed using MAFFT v7 and manually adjusted in BioEdit v7.0 [22,23]. The phylogenetic relationships were inferred based on whole plastomes by maximum likelihood (ML). The best nucleotide substitution model was chosen using jModelTest2 under the Akaike information criterion (AIC) [24]. ML analyses were conducted in RAxML v8.2.12 with a rapid bootstrap analysis (BS; 1000 replicates) and searching for the best-scoring tree simultaneously [25].

### 2.3. Divergence Time Estimation

Divergence times were estimated under relaxed lognormal clock model in BEAST v2.6.0 based on the whole plastome matrix [26]. Two fossils were applied: (1) the fossil *Styrax elegans* (56–47.8 Ma) was set as the root age with uniform distribution [27], and (2) the fossil *Halesia reticulata* (37.2–33.9 Ma) were set to the stem age of *Halesia* with uniform distribution [28]. We ran MCMC searches for 100 million generations and sampled every 5000 generations. After assessing the convergence using Tracer v.1.7 [29], the maximum clade credibility (MCC) tree was calculated by TreeAnnotator v.2.6.0 [30].

### 2.4. Occurrence Data Collection and Niche Evolution Analyses

The occurrence records of *Sinojackia* were collected from Global Biodiversity Information Facility (GBIF, https://doi.org/10.15468/dl.pws456, assessed on 20 May 2024), the Chinese Virtual Herbarium (CVH, http://www.cvh.ac.cn/, accessed on 21 May 2024), and the published species records [11,12,13,14]. R package “CoordinateCleaner” were used to remove coordinates with uncertainty >10 km, with equal longitude and latitude values, and those near biodiversity institutions and within seas [31]. Totally, there are 28 cleaned coordinates obtained (Appendix A).

Twenty-one environmental factors were compiled for each record. Nineteen bioclimatic variables and elevation layers were downloaded from WorldClim database v.2.1 (https://www.worldclim.org/). The potential evapotranspiration (PET) was extracted from the CGIAR-CSI website (https://csidotinfo.wordpress.com/). The 21 environmental variables were analyzed using phylogenetic principal component analysis (PPCA) implemented in R package “phytools” 2.3.0 [32]. Based on the first axis of the PPCA results, a run of 10 million generations for the 21 environmental variables was performed in BAMM and visualized in R package “BAMMtools” [33].

### 2.5. Potential Distribution Areas Prediction

Further, on the clean occurrence records of *Sinojackia*, we only reserved one occurrence record within 5 km to reduce the spatial autocorrelation. Finally, 15 eligible occurrences were used in species distribution model (SDM) analyses (Appendix A). Nineteen bioclimatic variables of the Last Glacial Maximum (LGM), near-current climate (representative of 1970−2000), and future climate (2090: average of 2081–2100) were downloaded from the WorldClim database v2.1 (https://www.worldclim.org/, accessed on 10 June 2024) at 2.5 arc-min spatial resolution. Shared socio-economic pathway (SSP) scenarios are global emissions scenarios developed by the Intergovernmental Panel on Climate Change (IPCC). The climate projections are based on emissions scenarios for the future. Three scenarios were used in the climate change projections: SSP1-2.6 assumes that warming stays below 2 °C, with net zero CO_2_ emissions reached by 2050; SSP2-4.5 assumes that warming reaches 2.7 °C by 2100; and SSP5-8.5 assumes CO_2_ emissions approximately double that of current levels by 2100 and warming reaches 5.2 °C by 2100 (https://www.ipcc.ch/). We chose the three climate scenarios (SSP1-2.6, SSP2-4.5, and SSP5-8.5) for subsequent niche modeling. In addition, the 2022 Human Footprint data (HFP) were downloaded from www.wcshumanfootprint.org (accessed on 10 June 2024), and was used to predict the potential distributions of *Sinojackia* species in the present and future. Only bioclimatic variables with Pearson’s |r| < 0.8 in each period were considered in the following analyses, and the relative importances of them were assessed using the jackknife method in MaxEnt v3.4.1 (https://biodiversityinformatics.amnh.org/open_source/maxent/, accessed on 10 June 2024). We calculated the potential distribution areas across the future after model building, and classified the results of SDM under four ranks (i.e., highly suitable, moderately suitable, less suitable, and unsuitable) using the “Reclassify tool” in MaxEnt. Finally, we calculated the area changes in the past and future compared with current potential distribution area using SDM toolbox v2.4 [34].

## 3. Results and Discussions

### 3.1. Phylogeny of Sinojackia

The maximum likelihood phylogeny based on the whole plastome data has supported the idea that *Sinojackia* is a monophyly with the highest supporting values (ML-BS = 100%, Figure 1). All species of *Sinojackia* was supported as a monophyletic with moderate to highest supporting values and *Sinojackia* was divided into two subclades (Figure 1). One subclade contained *S. microcarpa*, *S. sarocarpa*, and *S. huangmeiensis* with low supporting values (ML-BS < 50%), and the latter two species were grouped together with the highest supporting values (ML-BS = 100%). Another subclade also contained three species, *S. oblongicarpa*, *S. rehderiana* and *S. xylocarpa*, with highest supporting values (ML-BS = 100%), and the former two species were clustered together with moderate supporting values (ML-BS = 73%). Furthermore, our results supported the previous studies that *Changiostyrax dolichocarpus* was not a member of *Sinojackia* [17,18,19].

Previous studies based on several DNA markers did not resolve the phylogenetic relationships within the genus *Sinojackia* [17,18]. Our phylogenetic results have almost clarified the relationships within *Sinojackia*, which were similar with that of Jian et al. [19], with a difference in supporting values of a few nodes. However, the genus *Sinojackia* may experience rapid radiation during its evolutionary history, which implied that more nuclear data are needed to fully resolve their close phylogenetic relationships.

### 3.2. Divergence Time and Niche Evolution of Sinojackia

Our dated-phylogeny results showed that the genus *Sinojackia* originated at 12.61 Ma (95% HPD: 4.43–23.82), and diversified at 9.32 Ma (95% HPD: 2.51–18.68) (Figure 2A). These results were slightly later than the result of Jian et al. [19], and perhaps this difference contributed to the different calibration strategies. Furthermore, the six species diverged since the late Miocene (Figure 2A), which is consistent with Jian et al. (4.69–9.44 Ma) [19].

For the genus *Sinojackia*, the niche PPCA result showed that PC1 had a higher contribution from the aridity index (AI), temperature seasonality (Bio4), minimum temperature of coldest month (Bio6), and elevation (Table 1). The rate of niche evolution remained stable since the origin of this genus, and it experienced a slow increase toward the present (Figure 2B). The genus *Sinojackia* inhabited the Asian subtropical EBLFs [11,12,13,14,15]. Our ecological niche modeling analyses indicated that AI is the most important climatic variable for the niche evolution of *Sinojackia* (Table 1). The intensification of the Asian monsoon climate in the Late Miocene and Pliocene brought enough precipitation, which provided a suitable and stable environment for the divergence of *Sinojackia*. From 6 Ma onward, large-scale ice sheets in the Northern Hemisphere and the Icehouse state occurred [35], which had seriously affected the geographical distribution of plant lineages in the Asian EBLFs (Hai et al.) [36]. Undoubtably, frost is the key constraint for evergreen broad-leaved plants, as shown by our results, which indicates that the temperature seasonality and minimum temperature of the coldest month are important influential climatic variables for the *Sinojackia* species (Table 1). Meanwhile, a decrease in precipitation occurred in East Asia since the Pliocene [37]. Although a low temperature and decreased precipitation might have progressively deteriorated East Asian subtropical EBLFs since 6 Ma, *Sinojackia* is shrubs or trees and is not the dominant species in EBLFs. Only *S. henryi* sparsely inhabited forested slopes and ravines at 100–3500 m, while other species were distributed in a narrow altitude range [10,11,12,13,14], which implied that the elevation may play a major role in the niche evolution of *Sinojackia*. We, therefore, suggested that the dominant trees in Asian subtropical EBLFs might provide a suitable niche for the development of *Sinojackia*.

**Figure 2 biology-13-01085-f002:**
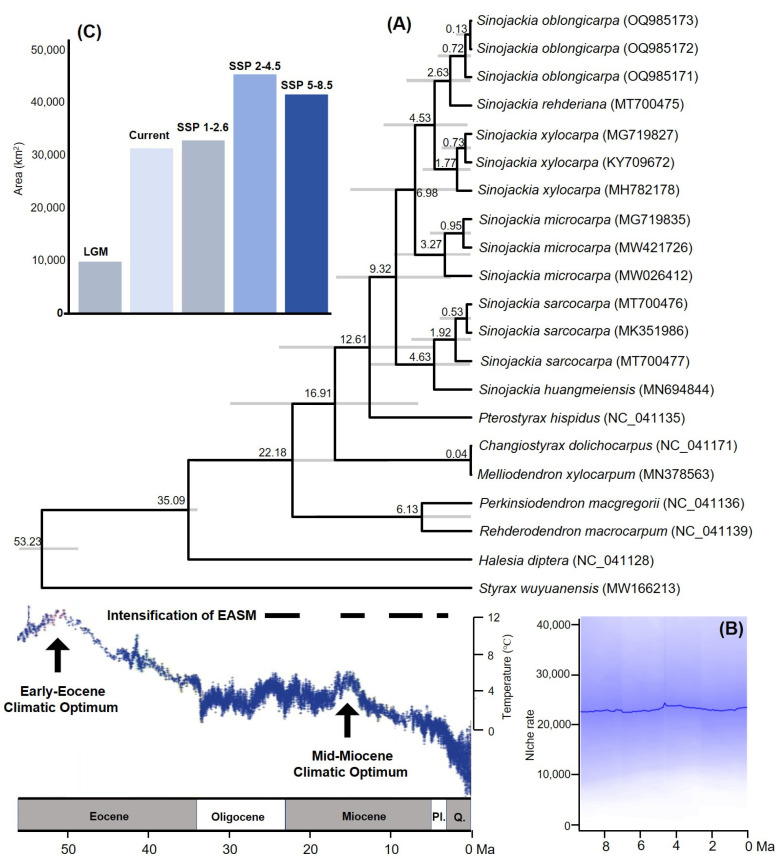
Dated-phylogeny (**A**), niche evolution (**B**), and potential distribution areas (**C**) of *Sinojackia*. EASM: East Asia Summer Monsoon, LGM: Last Glacial Maximum, and SSP: shared socio-economic pathway (SSP). Climate change was modified from Zachos et al. [38]. Pl.: Pliocene, and Q.: Quaternary.

### 3.3. Potential Distribution and Its Conservation

The AUC value for *Sinojackia* in the MaxEnt model was 0.999, indicating a high predictive performance. Moreover, the high suitability of the predicted near-current distributions were generally good representations of the observed distributions of *Sinojackia* (Figure 3). Our result of the MaxEnt model showed that the distribution of *Sinojackia* species is mainly impacted by the precipitation of the warmest quarter (Bio18) and the temperature seasonality (Bio4) during the LGM (Figure 3). The ancestral suitable geographical ranges of *Sinojackia* have changed since the LGM under the MaxEnt model. Compared with the near-current distribution areas (31,597.22 km^2^), the suitable distribution range of *Sinojackia* during the LGM (9982.64 km^2^) was constrained in eastern Asia. It is widely acknowledged that the precipitation and temperature also had been declining dramatically during the LGM [38,39]. In Asia, with the uplift of the Himalayas and Hengduan Mountains, Chinese flora had not been influenced extremely [40]. Lu et al. [10] investigated the spatio-temporal divergence patterns of Chinese flora and indicated that 66% of the angiosperm genera in China did not originate until early in the Miocene epoch, and proposed that eastern China had served as both a museum and a cradle for woody genera. Therefore, we deemed that *Sinojackia* had been relict in eastern Asia during the LGM after it originated in the middle Miocene.

Compared to the distribution range during the LGM (9982.64 km^2^), the current distribution areas of *Sinojackia* have expanded with a relatively higher temperature and more precipitation. Under the three climate scenarios in the future (2081–2100), the distribution range of *Sinojackia* becomes more fragmented (Figure 3), but the areas of the highly suitable range are expanded (Figure 2C), with the highest suitable range under SSP2-4.5 (33,107.64 km^2^ at SSP1-2.6, 45,625.00 km^2^ at SSP2-4.5, and 41,805.56 km^2^ at SSP5-5.8). In the current periods and the three future climate scenarios (SSP1-2.6, SSP2-4.5, and SSP5-8.5), the human footprint (Hfp) is the most significant variable influencing the distribution, and bio18 is also a main variable climatic factor (Figure 3). A previous study indicated that some narrow-ranging species may be positively influenced by the Hfp [41], although species with small ranges are regarded as more vulnerable to extinction [42,43].

The human footprint implies the human population pressure, land use, and so on [44]. There are two conflicting aspects related to the roles of the human footprint in the species’ suitable habitat. Most studies showed a negative relation between species conservation and the human footprint, because human activities such as land logging and livestock grazing could threaten the species’ suitable habitats [45,46,47]. However, Liu et al. [48] showed a positive relationship between the human population density and the density of scattered trees in China, which supported the idea that species conservation may benefit from human activities [46,48,49]. Our results also showed that, under human activities in 2081–2100, the highly suitable distribution of *Sinojackia* would expand. Therefore, we deemed that the positive human footprint on the habitat suitability of *Sinojackia* may be due to ecosystem conservation and protection management. Moreover, all *Sinojackia* species are listed as protected wild species in China due to their narrow distribution and high gardening values. Until now, more than 12 botanical gardens in the world transplant the *Sinojackia* species as ex situ conservation, and we emphasize that it is necessary to conduct more comprehensive field surveys of *Sinojackia* species for in situ conservation.

## 4. Conclusions

In this study, we reconstructed the phylogeny, estimated the divergence time, conducted niche evolution, and predicted the potential habitat area of *Sinojackia*, a Chinese endemic genus, and the main results showed the following: (1) The monophyly of *Sinojackia* was well-supported based on whole plastomes, and the interspecific relationships among the genus have been nearly resolved; (2) *Sinojackia* originated in the middle Miocene and diverged in the late Miocene; (3) the aridity index had the highest contribution for the niche of *Sinojackia*, and the niche evolution rate experienced a slow increase toward the present; (4) the main environmental variable affecting the distribution of *Sinojackia* is the precipitation of the warmest quarter in the LGM, while the human footprint is the main variable in the current era and 2081–2100; and, (5) compared to the highly suitable distribution area in the LGM, the genus *Sinojackia* would expand during the near-current era and 2081–2100, and human activities are positively related to this species’ conservation.

## Figures and Tables

**Figure 1 biology-13-01085-f001:**
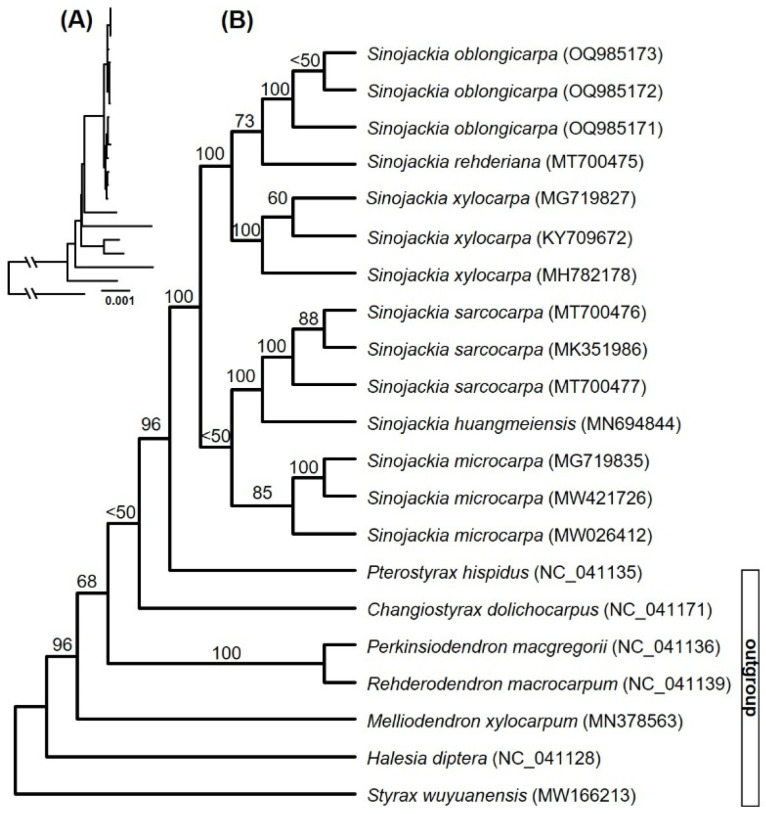
Maximum likelihood phylogeny of *Sinojackia* based on the whole chloroplast genomes: (**A**) phylogram of *Sinajackia*, and (**B**) cladogram of *Sinajackia* with supporting values above branches.

**Figure 3 biology-13-01085-f003:**
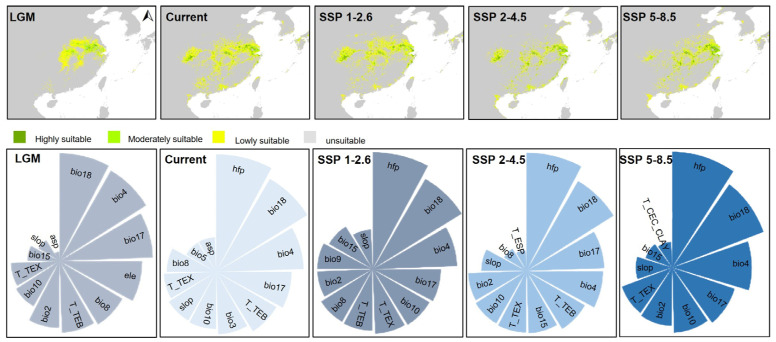
Potential distribution range of *Sinojackia* and their variables’ contributor from the LGM to the future climate scenario simulated by ecological niche models.

**Table 1 biology-13-01085-t001:** Niche PCA loadings of *Sinojackia*.

**Variable**	**Abbreviation**	**PC1 Loadings**
Annual mean temperature	Bio1	0.5733
Mean diurnal range (mean of monthly (max temp − min temp))	Bio2	−0.7832
Isothermality (Bio2/Bio7)	Bio3	0.7618
**Temperature seasonality**	**Bio4**	**−0.9409**
Max temperature of warmest month	Bio5	−0.5796
**Min temperature of coldest month**	**Bio6**	**0.9354**
Temperature annual range	Bio7	−0.8971
Mean temperature of wettest quarter	Bio8	−0.5424
Mean temperature of driest quarter	Bio9	0.4949
Mean temperature of warmest quarter	Bio10	−0.6924
Mean temperature of coldest quarter	Bio11	0.8818
Annual precipitation	Bio12	0.7670
Precipitation of wettest month	Bio13	0.8511
Precipitation of driest month	Bio14	−0.5485
Precipitation seasonality	Bio15	0.7265
Precipitation of wettest quarter	Bio16	0.9082
Precipitation of driest quarter	Bio17	−0.5669
Precipitation of warmest quarter	Bio18	0.7736
Precipitation of coldest quarter	Bio19	−0.5239
**Elevation**	**Ele**	**0.9178**
Potential evapotranspiration	PET	−0.8708
**Aridity index**	**AI**	**0.9999**

The variables with the largest contribution and the highest PC1 loading are indicated in bold.

## Data Availability

The original contributions presented in this study are included in the article/Appendix A. Further inquiries can be directed to the corresponding author.

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
