# Peer review of "Niche Evolution and Conservation of a Chinese Endemic Genus Sinojackia (Styracaceae)"

_biology, 2024, doi:10.3390/biology13121085_

Round 1
Reviewer 1 Report
Comments and Suggestions for Authors
Climate change and human activities are expected to have a profound impact on the distribution of species, especially for narrow distributed species. They first reconstructed the phylogenomic tree and estimated the divergence time of Sinojackia, and the results showed that this genus originated at middle Miocene and diversified since the late Miocene. Through aridity was contributed highest for its niche evolution, but the niche evolution rate increased slowly. Precipitation of warmest quarter was a main environmental variable affecting the distribution of Sinojackia in LGM, while human footprint is the main variable in near-current and 2081-2100. Interestingly, the human activities were positive related to the distribution expansion in future. This study provides potential areas for in situ conservation of Sinojackia, but also shed new insight into the role of human footprint in endemic species. However, there still are some concerns need to be considered:
1. How about the in situ and ex situ conservation of Sinojackia now?
2. Please provided the sampling information as supplementary material.
Author Response
Q1. How about the in situ and ex situ conservation of Sinojackia now?
Response: Thanks for your comment. We have added the conservation information in INTRODUCTION (Lines 54-58).
Q2. Please provided the sampling information as supplementary material.
Response: Revised as your comment (new Table S1).
Reviewer 2 Report
Comments and Suggestions for Authors
The manuscript titled “Niche evolution and conservation of a Chinese endemic genus Sinojackia (Styracaceae)”. This study reconstructed the phylogeny and evaluated the divergence time of Sinojackia and conducted the niche evolution and predicted the potential habitat area of Sinojackia to future habitat. The study succeeds in estimating the phylogentic divergence, highest factor for niche evolution, environmental variables and prediction of habit suitability in some future climatic scenarios. Overall, this research is potentially full of interest, as it addresses the relevant topic. I have few comments and suggestions.
Major issues
1. The authors only gathered data from GBIF and the Chinese Digital Herbarium, and I am unable to locate your field data points from the distribution range. However, the author's alteast should conduct extensive field surveys from a few notable Northwestern Provinces to augment the data for more accuracy and reliability. Furthermore, distribution data is often biased towards various geographic places that may or may not be freely accessible, as is well known.
2. I recommend analyzing and interpreting species range change of the studied of Sinojackia species for better understanding in terms of quantification of species loss or gain habitat in different climatic scenarios.
3. Elevation is reported have significant impact on the distribution of Sinojackia species in this work and ongoing climate change have significant impact on shifting elevation range of many reported species. However, the manuscript does not reflect much discussion and interpretation of this important parameter and needs consideration.
4. I recommend authors to add a brief and separate paragraph in introduction section about the habitat and some morphological features of various species of Sinojackia that will enrich readers with better understanding and with greater precision.
Minor issues
5. When using an acronym for the first time, authors are suggested to write the entire term first, then the abbreviation in parenthesis. For instance “LGM”.
Wish you luck
Author Response
Q1. The authors only gathered data from GBIF and the Chinese Digital Herbarium, and I am unable to locate your field data points from the distribution range. However, the author's alteast should conduct extensive field surveys from a few notable Northwestern Provinces to augment the data for more accuracy and reliability. Furthermore, distribution data is often biased towards various geographic places that may or may not be freely accessible, as is well known.
Response: We absolutely agreed with your opinion on distribution data. It is recorded that the Sinojackia maintained a small number of populations with very small population size (Yao et al., 2005, Biodivers. Sci. 2005, 13, 339-346; Jian et al., Diversity 2024, 165, 305). In this study, we collected the occurrences from different resources as possible as we can, including GBIF, CVH, and the literatures on new species, biogeography and conservation. Therefore, the distribution data here represented the distribution range of this genus.
Q2. I recommend analyzing and interpreting species range change of the studied of Sinojackia species for better understanding in terms of quantification of species loss or gain habitat in different climatic scenarios.
Response: We have revised as you comment (Lines 224, 233, 238).
Q3. Elevation is reported have significant impact on the distribution of Sinojackia species in this work and ongoing climate change have significant impact on shifting elevation range of many reported species. However, the manuscript does not reflect much discussion and interpretation of this important parameter and needs consideration.
Response: We have revised as you comment (Lines 205-208).
Q4. I recommend authors to add a brief and separate paragraph in introduction section about the habitat and some morphological features of various species of Sinojackia that will enrich readers with better understanding and with greater precision.
Response: Thanks for your comment. We have added the morphological and habitat information in INTRODUCTION (Lines 53-61).
Minor issues
Q5. When using an acronym for the first time, authors are suggested to write the entire term first, then the abbreviation in parenthesis. For instance “LGM”.
Response: Thanks for your comment. We have added the full name of EASM, LGM and SSP when they appeared at the first time.
Reviewer 3 Report
Comments and Suggestions for Authors
The article is devoted to the important topic of the impact of global warming on rare plants and the prospects for preserving valuable plant species. The authors examined data on the current distribution of Sinojackia species, determined the most favorable combination of climatic indicators for this genus, analyzed different climate change scenarios, and described probable changes in the highly suitable range. With global warming, the most suitable area may expand, so conservation of species in situ is possible.
In my opinion, one cannot expect that Sinojackia species (woody plants with a long life cycle) will quickly spread throughout the entire suitable range.
The article will be of interest to a wide range of readers.
A few minor notes:
1) abbreviations EASM (Figure 2), LGM, SSP are not deciphered
2) Methods, line 133: it is necessary to briefly describe the scenarios SSP1-2.6, SSP2-4.5 and SSP4-8.5 and provide a reference to the article where these terms are described in more detail
3) Figure 2, in figures B and C, units of measurement on the vertical axes are not indicated
4) Left graph at the bottom of figure 2 (temperature change from Eocene to Quaternary) - no source reference. For which region is the data given?
Author Response
Q1. 1) abbreviations EASM (Figure 2), LGM, SSP are not deciphered
Response: Thanks for your comment. We have added the full name of EASM, LGM and SSP in the legend of Figure 2.
Q2. 2) Methods, line 133: it is necessary to briefly describe the scenarios SSP1-2.6, SSP2-4.5 and SSP4-8.5 and provide a reference to the article where these terms are described in more detail
Response: We have described the three scenarios and provided the website of Intergovernmental Panel on Climate Change (IPCC). (Lines 138-152)
Q3. 3) Figure 2, in figures B and C, units of measurement on the vertical axes are not indicated
Response: Thanks for your comment. We have added the vertical axes in the updated Figure 2.
Q4. 4) Left graph at the bottom of figure 2 (temperature change from Eocene to Quaternary) - no source reference. For which region is the data given?
Response: Thanks for your comment. We have revised it.